# Regulation of Oxygen Tension as a Strategy to Control Chondrocytic Phenotype for Cartilage Tissue Engineering and Regeneration

**DOI:** 10.3390/bioengineering11030211

**Published:** 2024-02-23

**Authors:** Mikko J. Lammi, Chengjuan Qu

**Affiliations:** 1Department of Medical and Translational Biology, Umeå University, SE-90187 Umeå, Sweden; 2Department of Odontology, Umeå University, SE-90187 Umeå, Sweden; chengjuan.qu@umu.se

**Keywords:** cartilage, oxygen tension, tissue engineering

## Abstract

Cartilage defects and osteoarthritis are health problems which are major burdens on health care systems globally, especially in aging populations. Cartilage is a vulnerable tissue, which generally faces a progressive degenerative process when injured. This makes it the 11th most common cause of global disability. Conservative methods are used to treat the initial phases of the illness, while orthopedic management is the method used for more progressed phases. These include, for instance, arthroscopic shaving, microfracturing and mosaicplasty, and joint replacement as the final treatment. Cell-based implantation methods have also been developed. Despite reports of successful treatments, they often suffer from the non-optimal nature of chondrocyte phenotype in the repair tissue. Thus, improved strategies to control the phenotype of the regenerating cells are needed. Avascular tissue cartilage relies on diffusion for nutrients acquisition and the removal of metabolic waste products. A low oxygen content is also present in cartilage, and the chondrocytes are, in fact, well adapted to it. Therefore, this raises an idea that the regulation of oxygen tension could be a strategy to control the chondrocyte phenotype expression, important in cartilage tissue for regenerative purposes. This narrative review discusses the aspects related to oxygen tension in the metabolism and regulation of articular and growth plate chondrocytes and progenitor cell phenotypes, and the role of some microenvironmental factors as regulators of chondrocytes.

## 1. Introduction

The Global Burden of Disease 2019 study reported that more than 1.5 billion people suffer from musculoskeletal diseases, encompassing conditions like osteoarthritis (OA), rheumatoid arthritis, low back pain, fractures and other traumatic situations [1]. Cartilage injuries and OA are of significance to public health due to their high prevalence, impact on well-being and economic load. The global prevalence and burden of OA are estimated to increase further in future decades [2]. Importantly, OA ranks 11th as a leading cause of global disability [3], causing pain and socioeconomic costs, particularly in aging populations [4]. A study of the Skåne region in Sweden showed that, among individuals aged 45 and older, the percentage of doctor-diagnosed OA was 30.5% in women and 22.4% in men [5]. Post-traumatic injuries also often lead to progressive degenerative changes in articular cartilage, which are difficult to cure [6]. Throughout our daily routines, the motions of our body subject our bones and joints to significant mechanical stresses. These forces are mitigated and dispersed by the articular cartilage located at the extremities of the bones [7]. While the inherent design of articular cartilage enables nearly frictionless movement at the ends of bones, aging often leads to the gradual deterioration of the cartilage tissue. The process harms the typical molecular structure and arrangement of macromolecules within the normal cartilage framework [8]. Degeneration of the joint can also develop due to trauma, sports activity, obesity or occupational activities, which can lead to focal chondral or osteochondral defects [9].

The symptoms of OA encompass joint pain, swelling, stiffness and a restricted range of motion. While there is no definitive cure for OA, there are several treatment options available to manage its symptoms and improve patients’ quality of life. Today, conservative treatments, such as physical therapy and pain-alleviating OA drugs, are used as the first treatment for OA [10,11]. When the disease advances, orthopedic techniques are timely options [12]. Regenerative tissue-engineering approaches have also been investigated more and more frequently taking advantage of various types of synthetic or natural scaffolds and chondrocytes/stem cells, bioactive factors, exosomes and even gene therapies with, for instance, anabolic factors or non-coding RNAs [12].

## 2. Cartilage Repair Methods

Osteoarthritis is a troublesome disease to handle, given the limited regenerative capacity of articular cartilage, resulting in progressive defects in the affected joint [13]. The orthopedic choices to treat OA are surgical cell- and tissue-based methods or joint replacements [14]. Classically, articular cartilage defects have been treated by techniques such as joint irrigation, lavage, synovectomy, microfracture, mosaicplasty, and abrasion arthroplasty [15]. In advanced OA, total joint replacement surgery has been found to be an efficient treatment, based on experience accumulated over more than the past 50 years [16]. It is currently the standard treatment for end-stage OA [14,17]. Despite the mostly good clinical outcomes, up to 28% of patients having gone through a knee or hip arthroplasty may consider themselves dissatisfied with the result [18].

In addition, prostheses have a limited lifetime and certain other problems are related to them, for instance, wear and tear on the implant material leading to the release of potentially harmful particles [19]. Thus, the development of complementary therapies, possibly combined with the use of biomaterials, are intriguing, since these techniques would potentially build up a repair tissue of cartilage in a more physiologically natural manner [20]. At best, these applications would restore the cartilage with a repair tissue that closely resembles normal, biomechanically durable tissue.

The initial documentation of treating cartilage defects in the knee with human autologous chondrocyte implantation dates back to 1994 [21]. This technique requires two orthopedic operations, so that cartilage pieces can be collected from the marginal areas of the joint during first operation, then enzymatically released and expanded and finally implanted into the site of injury and sealed with a periosteal flap [21]. This method has further evolved to second- and third-generation autologous chondrocyte implantation techniques, exploiting porcine collagen type I and III membranes to replace the periosteal flap, and a hydrated scaffold wherein the chondrocytes are suspended, respectively [22]. An approach using minced articular cartilage has gained attention as well, which is cost effective since the operation can be completed in a single surgical session, without the requirement of chondrocyte expansion [23].

In a procedure known as cartilage mosaicplasty or an osteochondral autograft transfer system, cylindrical osteochondral grafts, ranging from 4 to 10 mm in diameter, are inserted into lesion holes within the compromised cartilage region [24]. A fibrin clot developing between the cylinders will subsequently generate fibrocartilage that connects the neighboring implanted cylinders. Challenges associated with this technique include the complexity of restoring the cartilage surface shape, the limited integration of implanted cylinders to the surrounding cartilage and the creation of new lesion sites as a result of tissue removal [24,25]. An osteochondral allograft technique by using fresh or off-the-shelf tissues has been exploited to avoid donor site morbidity [26,27].

The improved results of cell-based cartilage repair techniques have increased the heterogeneity of the patients eligible for cartilage repair, while the patient’s age, activity level, defect location and a delay in treatment are among the factors that influence the clinical outcome. Therefore, further studies are warranted to better understand how better tissue repair could be achieved. Since the cellular sources for chondrocyte implantations are scarce, the utilization of stem cells as an alternative for chondrocytes/cartilage have garnered significant attention. In particular, mesenchymal stem cells (MSCs) have raised interest since they can be isolated from a vast range of different tissues [28]. Globally, searching with the words “mesenchymal stem cells” resulted in 1599 hits for clinical trials at clinicaltrials.gov (accessed on 4 November 2023), with 141 hits for “MSCs and osteoarthritis”. The safety profiles for the use of MSCs have been good in general [29]. However, the clinical trials related to cartilage repair suffer, for instance, from the inclusion of a relatively low number of patients, non-standardized treatments regarding the source of MSCs, their cell isolation methods and cellular dosage [30]. Still, at least short-term clinical improvements were observed in a recent systematic review of MSC implantation outcomes in patients with knee OA [31].

Higher caution must be applied in the use of induced pluripotent stem cells, due to possible tumoricity, incomplete differentiation, immune rejection and genetic mutations or epigenetic changes [32]. Nevertheless, a search using the key words “induced pluripotent stem cells” on clinicaltrials.gov resulted in 137 hits. A review from the year 2022 identified 81 relevant observational and interventional clinical trials from 11 January 2021 performed in 15 different countries in North America, Europe, Asia and Australia [33]. Techniques were developed a while ago to differentiate embryonic stem cells and human induced pluripotent stem cells (hiPSCs) into chondrocytes [34,35,36].

## 3. Tissue Engineering of Cartilage

The general principle of tissue engineering is to generate a functional tissue, such as articular cartilage, to replace the damaged one in the body [37,38]. As the injected chondrocytes utilized for cartilage repair result in a soft implant, the duration for the repair tissue to generate a sufficient amount of extracellular matrix (ECM) is prolonged, leading to the delayed development of favorable biomechanical properties in the cartilage. Consequently, the recovery period for patients is extended. To address this, one approach involves cultivating cell-based tissue-engineered cartilage in a laboratory using scaffolds or scaffold-free systems, aiming to produce a transplantable construct [39]. Additionally, tissue-engineered biphasic osteochondral implants hold potential in replacing the tissues, which is essential for the mosaicplasty technique in cartilage repair. 

Strategies involving cell-based approaches for articular cartilage repair have primarily concentrated on either mature chondrocytes or adult MSCs [40], and even embryonic stem cells or hiPSCs can be differentiated into chondrocytes [36]. Utilizing mature chondrocytes would be a sensible option for cell-based reparative approaches, but there is a limited availability of donor sites, and the chondrocytes rapidly undergo a phenotype switch into dedifferentiated phenotypic state during cellular expansion [41], characterized by a progressively reduced expression of type II collagen and aggrecan. The alterations in phenotype are linked to a reorganized arrangement of actin filaments as the transition occurs from three-dimensional ECM to monolayer culture [42]. In addition, bone marrow stromal cells, or MSCs, have a multilineage capacity to generate chondrocytes, osteoblasts and adipocytes [43]. They also can assist chondrocytes in preserving their phenotype, but the possible phenotypic instability of implanted constructs [44] and factors like donor age and cell passage affecting the differentiation potential [45] are potential limitations of their use for clinical applications. In fact, the young age of transplanted cell recipients can improve the success of cartilage repair [46].

## 4. Environmental Factors as Inducers of Cartilage Formation

During chondrogenesis, a number of temporally regulated factors are involved, which guide the cartilage development, such as transforming growth factor β (TGF-β), bone morphogenic proteins, Wnt/β-catenin system, parathyroid hormone, Indian hedgehog, insulin-like growth factor and various fibroblast growth factors [47]. A central player in chondrogenesis is transcription factor sex-determining region Y-box 9 protein (Sox9), which has been shown to be essential for the initiation and perpetuation of chondrocytic phenotypes together with Sox5 and Sox6 [48], particularly inducing the genes encoding for type II collagen and aggrecan.

Articular cartilage exists within an intricate and ever-changing mechanical environment. This environment is marked by compressive and shear stresses along multiple axes, along with hydrostatic and osmotic pressures, which regulate chondrogenesis and the formation and maintenance of the cartilage matrix [47,49]. The anabolic and catabolic responses of the chondrocytes depend on the frequency, the mode (static vs. dynamic loading, compression, tension and both osmotic and hydrostatic pressure) and the duration of the loading [50,51]. Mechanical cues are also important for the chondrogenic differentiation of MSCs [52]. Various mechanosensors respond to mechanical stimuli, alterations in chondrocyte deformation and variations in cellular shape and volume to transmit them from ECM and pericellular matrix (PCM) into intracellular signals. Calcium channels are well-known ion channels involved cellular sensing. These include, among others, mechanosensitive ion channels known as Piezo channels [53], transient receptor potential cation channel subfamily V member 4 [54] and voltage-gated calcium channels [55].

Chondrogenesis is a complex process regulated by multiple mechanisms including growth factors, cytokines and physical cues [47]. The signaling molecules involved in chondrogenesis (mostly growth factors) have been widely studied to find out whether they would be beneficial for tissue engineering purposes [44]. Recently, exosomes and non-coding RNAs, microRNAs in particular, have also raised a lot of interest as regulators of cartilage maintenance and as players in OA development and progression, but also in cartilage regeneration [56,57]. 

Mechanotransduction also involves multiple pathways and mechanosensors regulating chondrocyte homeostasis. The thin PCM together with plasma membrane proteins play a crucial role in mechanotransduction together with cellular cytoskeleton [58,59]. The PCM has a specific organization which facilitates the transduction of biomechanical and biochemical signals [60]. The chondrocyte cytoskeleton has been recognized to be involved in chondrocyte morphology and ECM synthesis for a long time [41,61,62,63,64], although the mechanisms behind how cytoskeleton changes affect the cellular functions are still inadequately understood. Anyway, focal adhesion structures are changed by tension and cell shape, and the action of focal adhesions as mechanosensors depends on the tension balance between adhesion tension and cellular tension [65]. Mechanosensitive ion channels, such as Piezo1, can then induce calcium influx, potentially initiating many cellular pathways [66]. Talin is an actin-binding adaptor protein in focal adhesions, linking integrins to actin. Stretching and unfolding of talin are part of the force transmission system [67], which, under mechanical stress, can trigger, for instance, cytoskeletal restructuring, giving rise to integrin α_V_-mediated activation of anabolic TGF-β [68]. Other cytoskeletal components appear to have a role, since microtubule depolymerization by nocodazole inhibited dynamic loading-induced stimulation of proteoglycan synthesis [63]. 

Cytoskeleton and focal adhesions also contribute to MSC differentiation, so that chondrogenic differentiation is encouraged by the spherical morphology of the dispersed actin cytoskeleton [69]. The inhibition of the cytoskeleton organization-regulating RhoA/ROCK pathway was shown to increase chondrogenic gene expression, while reorganizing the actin cytoskeleton [70]. It also inhibited the dedifferentiation of articular chondrocytes [71], in line with the reported increase in aggrecan deposition and suppressed matrix metalloproteinase-2 production [72]. On the other hand, the reorganization of the actin cytoskeleton can activate downstream signaling cascades, such as Rho GTPases [73]. 

Microtubule-based primary cilium is a single, immobile cellular extension which participates in mechanotransduction as a part of the signaling pathways in chondrocytes [74]. It is a signal receiver which can initiate, for instance, downstream Hedgehog and Wnt signaling cascades [75]. It can transduce signaling generated by strain in the transcriptional machinery which involves Cited2, transcription factor Sp1 and hypoxia-inducible transcription factor-1α (HIF-1α), and extracellular ATP [76]. Recently, activation of Yes-associated protein (YAP), which is assumed to sense and respond to ECM stiffness, was also shown to inhibit inflammatory signaling and ECM degradation when primary cilia expression in primary chondrocytes was decreased [77].

## 5. Low Oxygen Effects on Chondrocytes and Chondrogenic Phenotype

Articular cartilage with a thickness of 2–4 mm thick is a tissue which has no blood vessels, nerves or lymphatics. Therefore, the chondrocytes have a limited supply of oxygen, and they mainly depend on anaerobic glycolysis in the deeper zones to maintain and repair the ECM using glucose as a fuel to generate ATP [78], while analyses of the rates of oxygen uptake and lactate production show that oxidative phosphorylation is responsible for generating less than 25% of the total ATP production [79]. Mathematical models have anticipated fluctuations in oxygen levels between 1–5%, so that the highest level of oxygen is found at the superficial zone of cartilage [80]. This is rather well in line with the measured ranges from 7% at the superficial layer to below 1% in the deep zone [81]. Since the oxygen tension in cartilage is lower than that in majority of the other tissues or organs in the body, this condition is physiological for chondrocytes, and the term physioxia has been used to describe it [82].

Despite the pronounced role of glycolysis in chondrocyte metabolism, mitochondria play a crucial role in chondrocytes’ physiology, for instance, by regulating redox homeostasis [83], its involvement in mitochondria-mediated apoptosis [84] and regulating cellular responses to stress and senescence [85,86].

In general, gene expressions under hypoxia are significantly regulated by HIFs, which, after induction by hypoxia, generally decline in 5 min to a level below even the basal one after returning to a normal oxygen atmosphere [87]. The HIF oxygen sensors depend on prolyl hydroxylation of HIF-1α at normoxia, leading to its ubiquitination and rapid proteasomal degradation after binding to the von Hippel–Lindau protein tumor suppressor protein (Figure 1) [88,89]. Generally, when hydroxylation decreases, the stabilized HIF-1α forms a dimer with HIF-1β and recruits additional co-activators. This protein complex then binds to hypoxia response elements (HREs) in the nucleus to activate a number of various genes [90]. A study collecting the HIF-1α target genes pointed out 98 genes and 20 cellular pathways that were involved, thus the responses to HIF-1α activation are manifold [91].

However, studies using infrapatellar fat pad stem cells from OA patients for chondrogenesis at low oxygen tension showed that it is actually HIF-2α, rather than HIF-1α, which is involved in improved chondrogenic response to low oxygen [92]. The response included the increased expression of the Sox5, Sox6 and Sox9 transcription factors [92]. Later, HIF-3α expression was also shown to be present in articular chondrocytes, while its expression was reduced in cells with a high expression of hypertrophic genes [93], suggesting it has a role as a regulator of chondrocyte terminal differentiation [93].

### 5.1. Low Oxygen Tension Effects on Chondrocytes of Articular Cartilage

Chondrocyte phenotype is accompanied by energy metabolism [94]. It has been shown that, under hypoxic conditions, the chondrocytes upregulate glucose transporter 1 expression and increase glucose uptake [95]. Hypoxia significantly enhances the deposition of ECM by the chondrocytes [96]. Under low oxygen tension (≤5% oxygen tension), stabilized HIF-1α and HIF-2α prompt the expressions of Sox9, aggrecan and type II collagen in comparison to cells cultured in 20% oxygen tension [97], and cause an increase in ECM deposition in epiphyseal chondrocytes (Figure 1) [98]. Further, three-dimensional encapsulation and hypoxia restored the chondrogenic phenotype of dedifferentiated chondrocytes [96], suggesting that low oxygen could be beneficial for cartilage tissue engineering. The expansion of rabbit articular chondrocytes at 5% oxygen was also adequate to improve chondrogenesis in chondrocytes aggregates after the expansion [99]. The need for nutrients and oxygen apparently varies depending on the culture system [100]. Thus, temporal adjustments of oxygen and nutrient levels are potentially valuable for the optimization of culture conditions, for instance, for tissue engineering purposes.

Most often, studies examining the impacts of low oxygen tension have been quite short in duration. However, it takes a rather long time for chondrocytes to grow and assemble cartilage tissue. Therefore, it is also crucial to understand how low oxygen tension affects the chondrocytes. Our laboratory has investigated how the long-term exposure (up to 28 days) of human chondrosarcoma cells to low oxygen in the presence of a ROCK inhibitor affect their proteome. Besides increasing the ECM secretion, a total of 44 proteins (16 up-regulated and 28 down-regulated ones) were observed using a quantitative label-free mass spectrometry analysis [101]. Certain hypoxia-associated proteins were among those, but proteins which were not previously associated with low oxygen tension were also present [101]. The ROCK inhibitor Y-27632 has been reported to prevent the dedifferentiation of human articular chondrocytes [71], increase aggrecan deposition and decrease matrix metalloproteinase-3 production [72]. Thus, it appeared to be interesting to study the long-term effects (up to 28 days) of the combined exposure to low oxygen tension and Y-27632 on human chondrosarcoma cells [102]. In fact, gene expressions of *Sox9*, *ACAN* and *COL2A1* were induced by the 5% oxygen tension at all time points investigated (2, 7 and 28 days), but in particular, *ACAN* and *COL2A1* were highly induced by simultaneous exposure to 5% oxygen tension and 10 μM of the ROCK inhibitor [102]. Analysis of secreted type II collagen and sulfated glycosaminoglycans supported this finding [102]. At a low oxygen tension, Y-27632 produced 64 up- and 37 down-regulated protein productions [102]. These include many proteins previously not associated with hypoxia or chondrogenesis, so their functional relevance to low oxygen responses needs further studies. The ROCK inhibitor also modified the expressions of a group of S100 proteins [102]. Some of the cellular processes related to hypoxia in the presence of ROCK-inhibitor are indicated in Figure 2.

S100 proteins are interesting considering cartilage functions. Comprising a substantial subfamily of proteins featuring the Ca^2+^-binding EF-hand, humans have over 20 small acidic S100 proteins, each weighing between 10 and 12 kDa [103]. The majority of S100 proteins have the capability to create homo- and heterodimers, with researchers identifying at least ten distinct combinations of heterodimeric S100 proteins [103]. Various members of the S100 protein family, along with alterations linked to diseased tissue, have been recognized in the articular cartilage of humans [104]. In the process of repairing defects on the articular surface, chondrogenic cells expressed the S100 protein (without specifying the particular S100 protein) at an early stage, preceding the initiation of the production of the typical cartilaginous matrix components [105]. As transcriptional targets of Sox trio, S100A1 and the S100B are located in late proliferative and prehypertrophic chondrocytes within the murine epiphyseal plate, and their silencing stimulated terminal chondrocyte differentiation [106]. In humans, they are homogeneously expressed in all zones of normal articular cartilage [107] but are lost in OA cartilage [108]. Moreover, the expression levels of *S100A1* and *S100B* are elevated in human articular chondrocytes in comparison to osteophytic ones [109]. These observations suggest a correlation between S100A1 and S100B and the mature chondrocyte phenotype. 

The induction of chondrogenesis-related genes in the presence of a ROCK inhibitor in hypoxia might offer new strategy for treating the cartilage injuries or OA. Low oxygen tension is a natural environment for chondrocytes; therefore, improved synthesis could be expected with the use of already approved ROCK inhibitor drugs such as fasudil, ripasudil and netarsudil.

### 5.2. Low Oxygen Tension Effects on Stem Cells or Progenitor Cells

Adult stem cell therapies for human cartilage diseases have raised significant interest, since specific cell culture factors have long been known to induce chondrogenesis in cell pellets in 3–4 weeks [110]. Since oxygen tension in cartilage ranges from 1–5%, it is logical to assume that low oxygen tension would guide the chondrogenic differentiation of MSCs. Indeed, cell survival, colony formation and the proliferation of rat bone marrow-derived MSCs were greater at 5% oxygen than at 20% oxygen [111]. Partly conflicting results on the benefits of low oxygen for chondrogenesis have been reported. One of the first studies investigating the effects of reduced oxygen tension showed that 2% oxygen tension inhibited both chondrogenesis and osteogenesis of adipose-derived MSCs [112]. In contrast, higher type II collagen and proteoglycan productions were induced by 2–5% oxygen in HIF-1α-dependent manner in rat bone marrow MSCs [113]. Cartilage-specific gene expressions also increased in human bone marrow-derived MSCs [114], and under chondrogenic culture conditions of alginate beads embedded with human adipose tissue-derived MSCs [115]. Some of these results clearly depend on whether the MSCs were expanded (1) at ambient 20% oxygen before exposure to hypoxia or (2) were permanently cultured in low oxygen conditions. Other factors can also affect chondrogenic differentiation, such as donor age, passage number and individual variability [116]. The expansion of MSCs has often been used to attain a sufficient cell count for further differentiation in cellular pellets. Measurements of the oxygen consumption rate in umbilical cord-derived MSCs showed approximately three-times-higher values at 21% oxygen compared with 1.5–5% oxygen [82]. Today, increasing evidence has been accumulated showing that hypoxic preconditioning of the MSCs is advantageous for their differentiation and regenerative potential [116,117,118,119,120,121]. 

It was noticed early on that the differentiation of MSCs frequently progresses to terminal differentiation, indicated by the presence of hypertrophic chondrocytes [122]. For tissue engineering and regeneration purposes, risks of graft instability and even transdifferentiation are undesired outcomes [123,124,125]. In bone marrow MSCs, low oxygen tension promoted chondrogenesis and appeared to inhibit hypertrophic differentiation in cell pellets and hydrogels [126,127], both important responses for cartilage tissue engineering approaches. Cultures of human adipose tissue-derived stem cells in elastin-like polypeptide resulted in a similar expressions of cartilage ECM molecules both with or without chondrogenic supplements, while 5% oxygen prevented the induction of type X collagen evident at 20% oxygen tension [128].

Pluripotent stem cells are also of interest in chondrogenic differentiation under hypoxic conditions. It has been reported that chondrogenically differentiated, self-assembled chondrocytes from embryonic stem cell embryoid bodies produced constructs with higher collagen production, increased proportion of type II and even better tensile and compressive moduli [129].

Although it remains to be seen whether hiPSCs are applicable in clinical human treatments, they can be valuable sources as disease models [130,131]. Therefore, it is important knowledge that hypoxia was shown to increase chondrogenic marker gene levels in hiPSCs, while inhibiting mesodermal marker genes, clearly improving the chondrogenic phenotype of the differentiated stem cells [132].

### 5.3. Low Oxygen Tension in Cartilage Tissue-Engineering

In general, tissue engineering’s goal is to create tissues which can be implanted clinically to replace failed tissues [133]. Although methods exist to use autologous osteochondral grafts [134] and expanded autologous chondrocytes [21] protocols avoiding the development of OA at the tissue donor site and possible problems related to the size of the defect are the current focus. Experiments aimed at cellular therapy have exploited chondrocytes or MSCs as cellular sources, variable types of scaffolds or self-assembling scaffold-free systems and many types of bioreactors or three-dimensional bioprinters [135]. Since the collection of MSCs does not harm the cartilage, they have been considered as a cell type with high potential for cartilage tissue engineering. 

The culture of clinical-size cartilage constructs leads to the inhibition of nutrient transport and oxygen diffusion in comparison to the cultivation of monolayer cultures or small cell pellets and creates oxygen gradients within the engineered tissues [20]. Factors affecting oxygen gradients depend, for instance, on the cell density and distribution, oxygen consumption rate and the oxygen diffusion rate in the construct, and scaffold porosity [136,137]. Both spatial and temporal variations in the oxygen content are obviously influenced by cellular content and cell metabolism [136]. The oxygen consumption rate also depends on the cell type, so that oxygen consumption was higher in MSCs than in fat pad-derived stem cells or chondrocytes at day 0 of the culture, adopting a rather similar uptake by day 25 [137]. In scaffold-free chondrocyte pellet cultures, even anoxia in high density cell pellets was predicted by a mathematical model, with the appearance of the necrotic core of the pellets being characterized by a lack of chondrogenic differentiation and matrix synthesis [138]. Immediate seeding of bovine articular chondrocytes at a high cellular density into transwell inserts after cell isolation and cultivation for up to 6 weeks at both 5% and 20% oxygen showed that constructs at 20% had better ECM production, while those cultured at 5% oxygen had a very low production of sulfated glycosaminoglycans in the middle part of the constructs [139].

It has been shown that MSC-seeded agarose-based engineered tissues have lower mechanical properties than those seeded with chondrocytes [140]. The zonal assembly of articular cartilage has an essential role due to its biomechanical properties. Although oxygen tension in larger constructs is a challenging variable to regulate, exploiting these gradients in construct cultures might be a fruitful basis on which to create cartilage constructs from MSC-seeded hydrogels with zonal gradients which resemble those in articular cartilage [141]. On the other hand, the selection of the cell type could be important since MSCs from the infrapatellar fat pad were better at supporting functional repair tissue than articular chondrocytes in cell-encapsulated agarose hydrogels [142].

The low availability of chondrocytes and stem cells, requiring invasive methods to isolate them for autologous clinical treatments, has raised interest in the conversion of cells like fibroblasts, tenocytes or dermis-isolated adult stem cells (DIAS) into chondrocytes [143]. Skin dermis would be a tempting source of cells due to its good healing capacity after sampling of the tissue sample. Further, chondroinduction of human dermal fibroblasts has been reported to occur in low oxygen conditions [144]. Hypoxic differentiation of DIAS was effective in increasing type II collagen and sulfated glycosaminoglycan production [145], showing the potential applicability of DIAS in the regeneration of at least fibrocartilage.

### 5.4. Low Oxygen Tension Related Extracellular Vesicles and Non-Coding RNAs

Recent studies have indicated that small bilayer membrane vesicles, generally named as extracellular vesicles (EVs), or exosomes secreted by, for instance MSCs, can provide a novel avenue for tissue repair and the treatment of various diseases [146]. Extracellular vesicles are secreted by almost every type of cell, and they can carry various types of contents, such as proteins, lipids and non-coding/coding nucleic acids, to their target cells through cell-to-cell communications. They are generated by different cellular processes, leading to subpopulations of EVs of varying size [147]. They are mainly categorized as exosomes (40–100 nm), microvesicles (50–1000 nm) and apoptotic bodies (500–2000 nm) [148]. Besides paracrine mechanism, the EVs can be targeted in the body by systemic delivery, and circulating exosomes can be isolated from biological fluids, such as blood, saliva and breast milk [149]. Non-coding RNAs, which have diverse interactions and form interconnected networks, have gained increasing interest as contents present in EVs [150]. Also, the use of EVs in drug administration and as diagnostic biomarkers has been found to have a great potential [151]. The paracrine activity used to transfer EV content is elemental in the response to disease [152].

There is not very much known about the possible EVs or non-coding RNAs associated with low oxygen tension. Preconditioning MSCs at low oxygen tension increased the release of secretome that induced the formation of blood vessel-like structures [153], or EVs with anti-inflammatory and angiogenic potential and microRNAs involved in the healing process [154]. Priming menstrual blood-derived stromal cells at physioxia (1–2% oxygen) modified the proteome of their EVs. Recently, a microarray analysis identified 162 long non-coding RNAs associated with low oxygen tension, of which LncZFHX2 and LncHOXA11 had the most consistent and uniform tendency to be expressed in quantitative RT-PCR analyses [155]. LcnZFHX2 knockdown up-regulated genes involved in ECM degradation, while down-regulating those of ECM components [155]. LncHOXA11 knockdown did not have significant effects on these genes or proteins [155].

Knowledge about MSC and chondrocyte secretomes at hypoxia is still limited due to a lack of research, and novel findings which could be useful for regenerative purposes are likely to be revealed in the future.

## 6. Conclusions

Low oxygen tension, or physioxia, is an elementary environment for chondrocytes and chondrogenically differentiating MSCs and an increasing number of studies have shown evidence of its positive effects on chondrocyte differentiation and maintenance of chondrogenic phenotype. Priming chondrocytes and MSCs related to low oxygen tension may affect the cell-based treatments of damaged cartilage.

## Figures and Tables

**Figure 1 bioengineering-11-00211-f001:**
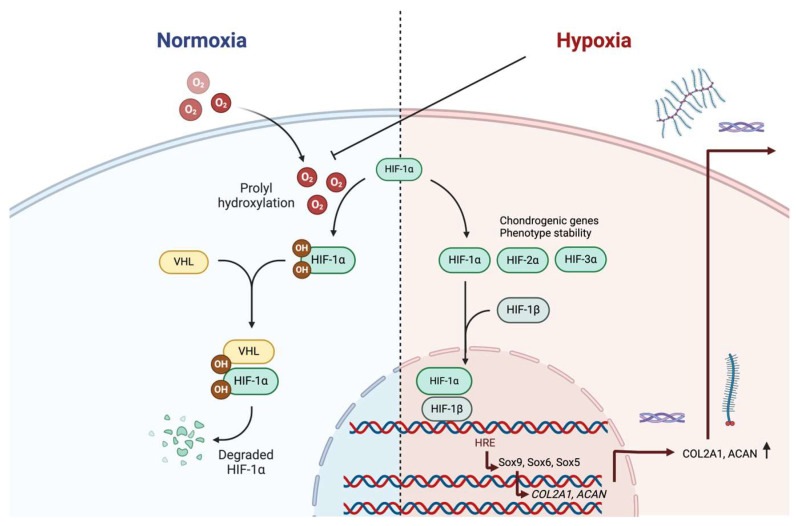
Hypoxia activation of HIF-1 and ECM production. Under normal oxygen tension, hypoxia-related genes are not activated, since transcription factor HIF-1α will be rapidly degraded in proteasomes after its prolyl hydroxylation and binding to von Hippel-Lindau protein (VHL). In low oxygen conditions, HIF-1α and HIF-1β bind together with hypoxia response element (HRE) and activate Sox9 and co-factors Sox5 and Sox6. Induction of chondrogenesis-related genes HIF-2α involves activation of Sox5, Sox6 and Sox9, and HIF-3α associates with stable chondrocyte phenotype. These will lead to increased gene expressions of procollagen α_1_(2) (COL2A1) and aggrecan (ACAN), and their secretion. Figure was created in BioRender.com.

**Figure 2 bioengineering-11-00211-f002:**
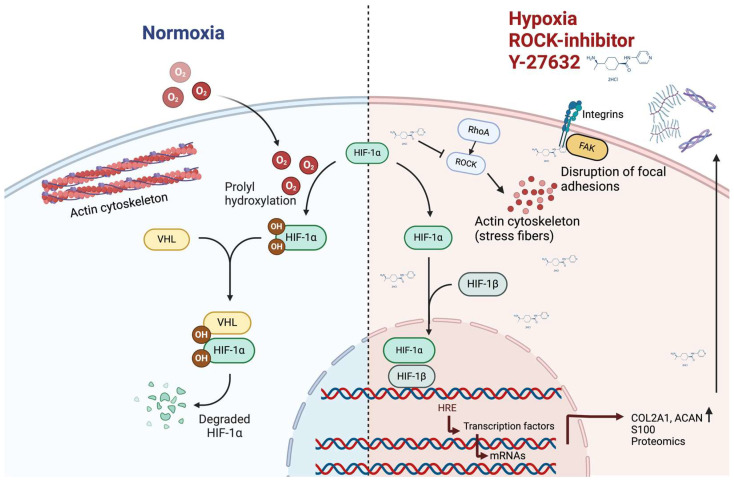
Long-term (4 weeks) 5% oxygen in the presence of ROCK inhibitor Y-27632 strengthened the expression of chondrogenesis-related genes ACAN and COL2A1 in chondrosarcoma cells. Numerous proteomic changes were also identified, for instance in S100A1 and S100B proteins. Targets for ROCK inhibitor include actin cytoskeleton, focal adhesions and their integrins. Abbreviations: hypoxia-inducible factor, HIF; von Hippel–Lindau protein, VHL; aggrecan, ACAN, type II collagen, COL2A1; focal adhesion kinase, FAK; Ras homolog family member A, RhoA, Rho associated coiled-coil containing protein kinase, ROCK. Figure was created in BioRender.com.

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
