# Peer review of "Regulation of Oxygen Tension as a Strategy to Control Chondrocytic Phenotype for Cartilage Tissue Engineering and Regeneration"

_bioengineering, 2024, doi:10.3390/bioengineering11030211_

Round 1
Reviewer 1 Report (Previous Reviewer 1)
Comments and Suggestions for Authors
Thank you for the thorough response to the comments of mine and all the other reviewers.
Comments on the Quality of English LanguageThis is generally fine. There are a few minor wording issues (lines 101-103, "Globally, search with the words ..." can be replaced with "Globally, searching with the words ..." or an alternative)
Author Response
We thank for the positive response. The sentence starting "Globally, search with words ..." has been corrected according to suggestion given by the reviewer.
Reviewer 2 Report (Previous Reviewer 2)
Comments and Suggestions for Authors
The manuscript is clear and addresses an important topic.
Line 18: "acquisition" is misspelled
Line 50: There are no DMOADs currently available.
Comments on the Quality of English Language
Well written.
Author Response
We thank for the positive response. We also thank the reviewer for spotting the typo in the word "acquisition", which is now corrected. We can also agree that at present there are no DMOADs available (although some publications occasionally like to state). We have now removed the word "disease-modifying".
This manuscript is a resubmission of an earlier submission. The following is a list of the peer review reports and author responses from that submission.
Round 1
Reviewer 1 Report
Comments and Suggestions for Authors
General Comments: This is an interesting review on the potential role of low oxygen tension in priming chondrocytes for cartilage repair treatments. It is generally well-organized and written. There is an extensive review of the effects of oxygen tension on chondrocytes and chondroprogenitor cells of a variety of types. This review should be of general interest to researchers.
There are a few general issues to clarify that will help identify the purpose and scope of the review. There are also more specific issues related to coverage of areas peripheral to the review, listed further below. There are a few typos and grammatical errors.
General Comments.
1. Abstract and Study Design. It would be useful to state, perhaps at the end, that the paper is a review and also the type of review. For example, was a systematic literature search done, at least on the oxygen aspect ? if so, how ?
2. Abstract and Student Design. It would also be useful to describe what aspect of oxygen tension regulation of chondrocyte phenotype in terms of what type of chondrocyte (eg articular/epiphyseal or physeal), and/or from what progenitor cell stage. This is important because the type of cell being affected may be quite different, as would be its microenvironment including matrix, mechanical, and chemical regulators..
3. In Section 6. Conclusion, it is stated. “We can expect that correct types of priming of chondrocytes and MSCs related to, for instance, low oxygen tension are likely to improve the therapeutic outcome of cartilage.” It would be better to temper this statement to something like “Priming of chondrocytes and MSCs related to low oxygen tension may affect the cell-based treatment of damaged cartilage.”
4. A summary figure would be helpful to provide an integrated view of the possible mechanisms and consequences of low oxygen regulation of chondrocytes.
Specific Comments
1. Section 2. Cartilage Repair Methods.
1.1. At the beginning of this section, one or several reviews or book chapters on the topic could be referred to.
1.2. Also, in this section, the approach of Osteochondral Allograft Transplantation should be added, perhaps after mosaicplasty/OATS, since it has distinguishing features from the latter.
2. Section 3. Tissue Engineering of Cartilage.
2.1. At the beginning of this section, one or several reviews or book chapters on the topic could be referred to.
Typos
T.1. In 5.2 Low oxygen tension … Paragraph 1 line 3, “... it is logic …” should be “ … it is logical …”
T.2. In 6. Conclusions “was shown to be involved in” should be deleted
T.3. In Abbreviations, the long form of ECM is missing the x in matrix
Comments on the Quality of English Language
A word processing grammar/spelling checker should catch some of the typos.
Reviewer 2 Report
Comments and Suggestions for Authors
This study reviews the role of oxygen tension in cartilage repair. Overall this is a timely and valuable manuscript. The following issues need to be addressed:
1. "The orthopedic choices to treat OA are surgical cell- and tissue-based mthods or joint replacements.". While we would love for this to be true, there are no surgical cell- and tissue-based treatments available that affect OA. Only joint replacement is available at this time for OA. This should be distinguished from cartilage defect repair procedures.
2. Overall the literature is not completely covered on the role of hypoxia in cartilage tissue engineering. There are many references that do not relate to this topic per se. Rather it would be helpful to include some additional references on difference stem cell types such as adipose stem cells, dermal stem cells, etc. where hypoxia has been shown to enhance chondrogenesis. See work by authors such as Athanasiou, Caplan, Guilak and others. A search on "cartilage" "chondrogenesis" and "oxygen" will show many of these.
3. Similarly, some studies have used hypoxia with iPSCs for chondrogenesis.
4. The paper would benefit from a figure and possibly a table summarizing the different studies that have examined oxygen effects on chondrogenesis.
5. There is a lot of background on clinical methods for cartilage tissue engineering but this does not relate much directly to the topic of the review.
Reviewer 3 Report
Comments and Suggestions for Authors
The manuscript provides a clear overview of the significant challenges posed by cartilage defects and osteoarthritis, particularly in aging populations and global healthcare systems. It briefly outlines treatment methods but lacks depth and supporting evidence. The concept of regulating chondrocyte phenotype through oxygen tension is introduced. Strengthening the manuscript could be achieved by structuring it more effectively with headings and subheadings and providing a more in-depth discussion on the potential of oxygen tension regulation. In conclusion, the manuscript holds promise, further enhancing its impact and scientific rigor.
Comments on the Quality of English LanguageIt is well written..
Reviewer 4 Report
Comments and Suggestions for Authors
This is a concise review of current work in engineering cartilage using several cell sources. Some comments that should be addressed.
1. On pg. 2 There is a statement in which 'stem' is in brackets, it is unclear what the authors are implying by this. Are they trying to indicate that they may not be stem cells?
Pg.2/3 The authors should include in their discussion which potential therapies are actually approved and in which country. This is especially important when discussing iPSC based therapies.
On pg. 4 first full paragraph the sentence that begins "The condrocyte cytoskeleton.... during mechanical loading." It is unclear how the cytoskeleton would affect the expression of ECM proteins. The authors should explain this more thoroughly.
On pg.6 section 5.2 the authors state the co-culture in low oxygen the articular chondrocytes "protect the hypertrophy of MSCs". Again how does this happen is it low oxygen specific or does this occur in co-culture regardless of the oxygen tension.
Comments on the Quality of English LanguageGenerally the English is fine, there are places where the tense needs to be fixed or articles are missing that would make the writing flow better.